# An immunofluorescence microscopy assay to discriminate distinct expression patterns of HIV-1 Gag and Nef proteins in HIV-1 provirus-harboring cells

**Rosana Wiscovitch-Russo**[1◉]**, Hongyan Sui**[1◉*]**, Mindy Smith**[2]**, Hiromi Imamichi**[2]**, H Clifford Lane**[2]**, Tomozumi Imamichi**[1]

**1** Laboratory of Human Retrovirology and Immunoinformatics, Applied and Developmental Research Directorate, Frederick National Laboratory for Cancer Research, Frederick, Maryland, United States of America, **2** Laboratory of Immunoregulation, National Institute of Allergy and Infectious Diseases, National Institutes of Health, Bethesda, Maryland, United States of America

◉ These authors contributed equally to this work.
* hongyan.sui@nih.gov

## Abstract

Over 95% of HIV-1 proviruses are defective and were once considered clinically irrelevant. However, growing evidence shows that these defective proviruses can still be transcribed and translated into viral proteins. Here, we developed an improved immunofluorescence protocol that combines two anti-Nef antibodies with one anti-Gag antibody, along with membrane and nuclear staining, enabling direct visualization of protein expression and localization. This method allows detailed characterization of the expression patterns and subcellular distribution of Gag and Nef proteins derived from defective proviruses. The protocol provides a practical tool for investigating the potential functions of proteins expressed from defective HIV-1 proviruses and for facilitating the ability to determine the biologic activity of cells harboring defective HIV-1 proviruses in patients living with HIV.

## Introduction

Human immunodeficiency virus (HIV) is a retrovirus that establishes lifelong infection by integrating its genome into host DNA [1]. This process requires reverse transcriptase (RT), a viral enzyme that converts the single-stranded viral RNA genome into double-stranded viral DNA. Because RT lacks proofreading activity and exhibits low fidelity, errors frequently occur during reverse transcription, resulting in deletions, insertions, or nucleotide substitutions within the viral DNA [2]. The integrated viral genome, termed a provirus, can remain transcriptionally silent in a latent or dormant state, yet retains the ability to reactivate and produce infectious viruses [3,4]. Although antiretroviral therapy (ART) suppresses viral replication to clinically undetectable levels, HIV-1 persists in CD4 + T cells in a latent form not removed by the

**Data availability statement:** All relevant data are within the manuscript and its Supporting information files.

**Funding:** This project has been funded in whole or in part with federal funds from the National Cancer Institute, National Institutes of Health, under contract number HHSN261200800001E.

**Competing interests:** The authors have declared that no competing interests exist.

immune system or ART. This latent reservoir is a major barrier to a cure [5]. Defective forms of HIV provirus are commonly detected in individuals receiving long-term ART. It is estimated that greater than 95% of HIV-1 proviruses persisting in peripheral blood mononuclear cells (PBMCs) are defective [4,5]. While intact proviruses are capable of replicating and assembling infectious viruses, defective proviruses contain deletions, insertions, or nucleotide substitutions in their genomes rendering them incapable of producing intact, replication competent viruses. Nevertheless, some defective proviruses remain transcriptionally active and capable of expressing viral proteins [4,6]. Viral proteins produced by defective proviruses could be involved in HIV pathogenetic effects and lead to persistent inflammation as has been reported in people living with HIV-1 despite effective ART [7–10]. Therefore, further investigation into the expression patterns of specific HIV proteins in cells will provide valuable insights into the pathogenesis of defective HIV proviruses.

Gag and Nef are essential for HIV replication and immune evasion. Gag is a major structural protein essential for virus assembly and maturation, while Nef is a multifunctional accessory protein known for downregulating cell surface markers (e.g., CD4 and MHC-I) thus reducing recognition by cytotoxic T lymphocytes [11–13]. More importantly, Nef facilitates trafficking of Gag to the plasma membrane, thereby enhancing cell-to-cell transfer of mature HIV-1 virions [14–17]. Although defective HIV-1 proviruses are considered non-infectious, Western blot analyses of cells harboring defective proviruses have revealed detectable expression of Gag and Nef proteins [18]. However, further characterization of proteins translated from defective proviruses is still lacking, and the synergistic effects of these proteins remain unexamined. Therefore, in this study, we aimed to develop an immunofluorescence assay to help fill this critical gap in knowledge.

## Materials and methods

The protocol described in this peer-reviewed article is published on protocols.io (https://dx.doi.org/10.17504/protocols.io.4r3l21xqjg1y/v1) and is included for printing purposes as S1 File.

### Cells and reagents

H9 cells were purchased from the American Type Culture Collection (ATCC, Cat# HTB-176) and the H9MN (H9/HTLV-III$_{MN}$ NIH 1984) cells (BEI Resources, Cat# ARP-402) harboring a mixed population of defective and intact HIV proviruses [19,20] were from the NIH AIDS reagent program. Cell lines harboring a single clone of infectious or defective proviruses were isolated from H9MN: the H9MN-FI cells harbor a full-length intact (FI) HIV-1 provirus and the H9MN-FD cells contain a full-length defective (FD) provirus with a 1-bp inserted frameshift [18]. H9 cells and H9MN-FI cells served as negative and positive controls, H9MN-FD and H9MN cells were used as experimental samples to validate the established immunofluorescence assay. All cells were grown in 25 cm$^2$ flasks using RPMI medium supplemented with 10% fetal bovine serum (FBS), 25 mM HEPES buffer, and 10 µg/mL gentamicin. The cells were incubated at 37°C and 5% $CO_2$ with saturating humidity for 2–3 days until cells reached confluency.

## Gag and Nef antibodies

A total of 12 commercially available anti-Nef antibodies and 4 different anti-Gag antibodies were screened in this study, and all related information is provided in S1 Table.

## Imaging and analysis of confocal microscopy

Imaging was carried out on a Zeiss Axio Observer.Z1 motorized LSM800 confocal microscope using a Plan-Apochromat 63x/1.40 oil objective (Zeiss, Oberkochen, Germany). The images were acquired and processed in the ZEN imaging software (version 3.1, Zeiss) [21]. Colocalization analysis of the images were carried out using Fiji/Image J (2.16.0/1.54p) using the colocalization threshold plugin [22]. Channels were background-subtracted. The analysis was run with automatic thresholding (Costes method) to identify the overlapping signal region. Colocalization was quantified using Rcoloc, the Pearson's correlation coefficient calculated only within thresholded overlapping pixels/voxels. This metric reflects the degree of linear correlation between the two fluorophores specifically in regions where both are simultaneously detected. Global PCC and Mander's coefficients (M1, M2) were also generated by the software but were not used in the primary analysis unless otherwise indicated. Each colocalization analysis was performed independently using three different confocal images. The Rcoloc values are presented as mean±SD (n=3), and a Student's t-test was used to calculate the p-value to assess statistical significance.

## Expected results

To precisely define the plasma membrane area of the cells, Wheat Germ Agglutinin (WGA) staining was incorporated into the protocol. WGA is a lectin that binds to specific carbohydrate residues, primarily N-acetyl-D-glucosamine and sialic acid, found on the surface of a wide variety of cell membranes [23]. This binding allows one to highlight cell boundaries and study membrane protein localization. WGA staining following fixation with 4% formaldehyde led to strong intracellular staining, with slightly weaker staining of the plasma membrane boundary in both H9 and H9MN cells (Fig 1A). To minimize cytoplasmic diffusion of residual WGA fluorescent dye, the staining procedure was optimized by performing WGA staining prior to cell fixation. Cells treated with WGA prior to formaldehyde fixation showed more selective plasma membrane labeling, characterized by a thicker membrane signal and minimal intracellular staining (Fig 1B). Consequently, staining WGA on the plasma membrane prior to cell fixation was found to minimize the translocation of this broad-binding lectin into the cytoplasm, thereby reducing nonspecific staining of the intracellular components.

After screening 12 different anti-Nef and 4 different anti-Gag antibodies, we identified two anti-Nef antibodies and one anti-Gag antibody that performed well in the assay. The primary antibodies chosen for this study were mouse monoclonal anti-HIV-1 Nef clone EH1 (NIH-ARP, Cat# 3689) [24] and clone 3D12 (Thermo Fisher Scientific, Cat# MA1−71501), along with rabbit polyclonal anti-HIV-1 p24 (LS Bio, Cat# LS-C486990). The H9MN-FI cells harbor a single clone of intact HIV provirus, for this purpose the H9MN-FI cells serve as a positive control to establish and develop the immunofluorescence assay. In H9MN-FI cells, the EH1 clone of the anti-Nef antibody recognized the total endogenous Nef protein, but the strongest fluorescent signal was predominantly observed in the cytoplasm, with low co-localization with the Gag protein detected by anti-Gag antibody (Fig 2A). In contrast, the Nef protein detected by the 3D12 clone of the anti-Nef antibody showed a higher degree of co-localization with the Gag protein detected by the anti-Gag antibody and were predominantly localized at the plasma membrane of H9MN-FI cells (Fig 2B). This was made more evident by the WGA staining, as the Nef (3D12) and Gag antibodies were co-localized in regions of the plasma membrane labeled by the lectin. More importantly, the colocalization analysis demonstrated the correlation of the subcellular localization of Nef and Gag. For instance, Nef (EH1) and Gag showed a weak correlation coefficient (Rcoloc=0.4483±0.0775) indicating distinct subcellular localization of the Nef (detected by EH1 clone) and Gag (Fig 2C), whereas Nef (detected by 3D12 clone) and Gag showed a strong correlation coefficient (Rcoloc=0.7439±0.0948) supporting the observed co-localization of these two proteins

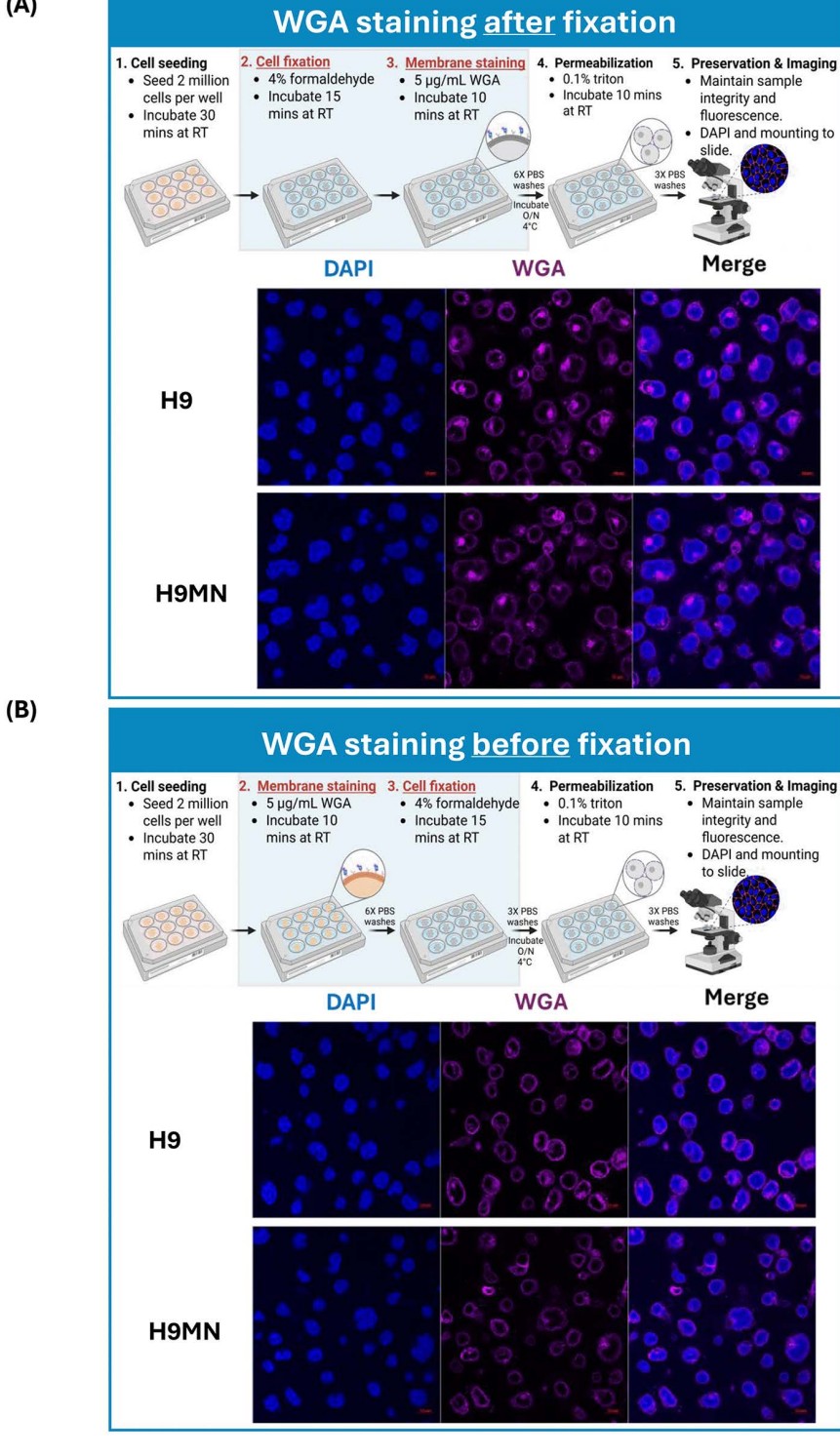

**Fig 1. Workflow and confocal microscopy images comparing WGA membrane staining before vs. after fixation.** Each microscopy image is paired with a schematic illustrating the WGA membrane staining approach. Nuclei were counterstained with DAPI (blue), and cell membranes and organelles were labeled with WGA conjugated to Alexa Fluor 647 (purple). Scale bar: 10 μm. **A)** WGA staining after fixation showed increased cytoplasmic staining. **B)** WGA staining before fixation showed clear definition of the plasma membrane, with minimal cytoplasmic staining.

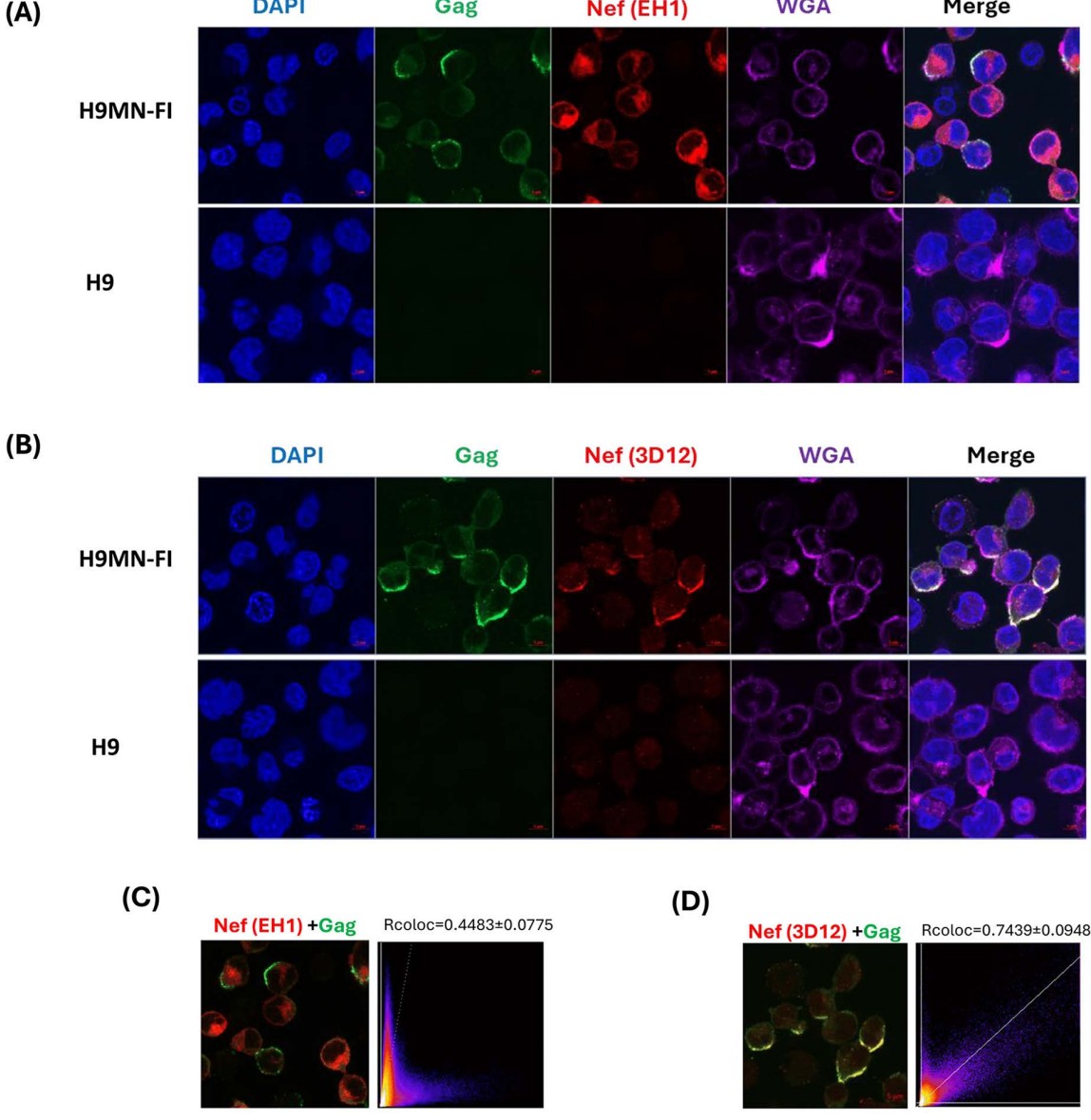

**Fig 2. Confocal microscopy images comparing Gag and Nef (clones EH1 and 3D12) expression in control samples. A & B)** For the confocal images, the stained nuclei (blue) and the plasma membrane (purple) were used to identify the cytoplasmic and membrane region, thus enabling the localization of Gag (green) and Nef (red) protein expression within the cells. Scale bar: 5 µm. **A)** Dual screening with Nef (EH1) and Gag antibodies. **B)** Dual screening with Nef (3D12) and Gag antibodies. **C)** Nef (EH1) and Gag antibodies showed a less correlation coefficient, while **D)** Nef (3D12) and Gag antibodies showed a high correlation coefficient. Colocalization analyses were performed across the entire regions shown in the images, using three independent confocal images. The colocalization coefficients (Rcoloc) are presented as mean±SD (n=3). Statistical significance was assessed using a student's t-test (p=0.0151).

([Fig 2D]). In both detection settings, the selected anti-Gag antibody consistently detected Gag expression localized to the plasma membrane in H9MN-FI cells. H9 cells were included as a negative control, and the results confirmed low background staining with both antibody sets. Overall, the chosen antibodies demonstrated consistent and specific localized staining patterns in H9MN-FI cells, validating their use for subsequent imaging analyses.

After the staining parameters were established using the positive and negative controls, we evaluated the expression patterns of Gag and Nef proteins in two H9 cells harboring distinct populations of HIV proviral clones. Specifically, we utilized the H9MN cell line, which harbors a mixed population of intact and defective HIV-1 proviruses, and the H9MN-FD cell line, which contains a single defective HIV-1 proviral clone [18]. Each assay included at least one positive and one negative control to ensure consistency and validate the staining patterns using both sets of antibodies—Nef (EH1) and Gag antibodies (1st and 2nd panel of Fig 3A) or Nef (3D12) and Gag antibodies (1st and 2nd panel of Fig 3B). In H9MN cells, polarized and membrane-localized signals were observed with both anti-Gag and anti-Nef (3D12) staining (3rd panel of Fig 3B), whereas the EH1 clone of the anti-Nef antibody detected primarily cytoplasmic Nef expression (3rd panel of Fig 3A). This observation is consistent with the characterization of H9MN cells, which harbor a mixed population of intact and defective proviruses. The detection of polarized membrane-localized Gag and Nef (3D12) signals likely reflects expression from intact proviruses, while the cytoplasmic localization of anti-Nef antibody EH1 clone may be associated with the expression from both intact and defective proviruses. By contrast, images from H9MN-FD cells show faint fluorescent signals for cytoplasmic-localized Nef (EH1) and Gag expression (4th panel of Fig 3A), with no detectable signal from the anti-Nef antibody 3D12 clone (4th panel of Fig 3B). The H9MN-FD cells did not exhibit membrane-localized Nef expression. This finding suggests that some defective HIV-1 proviruses are capable of translating proteins; however, the proteins were not in the same form as those translated from intact HIV provirus. Notably, the anti-HIV-1 Nef clone 3D12 targets conserved epitope in the Nef N-terminal close to the myristoyl group and myristoylation of Nef protein is essential for membrane association [25,26]. For the defective provirus in H9MN-FD, it is plausible that the 1-bp insertion in the RT gene may or may not be directly involved in the observed changes in Nef protein myristoylation and localization. The mechanisms underlying defective provirus Nef protein localization warrant further investigation. We found that using two anti-Nef antibodies, the EH1 and 3D12 clones, provides a valuable tool for identifying differences in Nef protein localization.

## Conclusions

In our protocol, two anti-Nef antibodies were selected for their ability to resolve subcellular localization of the protein. The 3D12 anti-Nef antibody specifically recognized Nef localized at the plasma membrane, whereas the EH1 anti-Nef antibody detected total Nef expression but predominantly highlighted its cytoplasmic localization. In the current study, the immunofluorescence assay was further optimized with WGA and nuclear staining, enabling clear distinction of the nuclear and the plasma membrane boundaries. The optimized method enabled identification of Gag and Nef protein localization, both essential for viral replication and infection. Nef's N- and C-terminal regions are partially conserved but can vary among HIV-1 subtypes and isolates. The 3D12 epitope (35−50 residues) is located within the N-terminal domain, while the EH1 epitope (194−206 residues) is located at the end of the C-terminal domain of Nef. However, amino acid changes in these regions may disrupt antibody binding. This assay was developed using cell lines carrying one of the most widely studied and prevalent HIV-1 variants (Group M, Subtype B). Nevertheless, additional studies are required to evaluate the cross-subtype recognition of the selected Gag and Nef antibodies across diverse HIV-1 subtypes.

In current established assay, both anti-Nef antibodies (3D12 and EH1) are mouse IgG1, so with the currently available reagents it is not possible to distinguish them using standard secondary antibodies. In the future, if clones from different species or IgG isotypes become available, the assay could be adapted to allow simultaneous detection of Nef and Gag on the same slide, which would be more convenient for clinical applications. Although the current assay has certain limitations, it still provides valuable insights into the composition of the viral reservoir in clinical samples: cells expressing Nef (detected by 3D12) are likely enriched for intact proviruses, whereas cells lacking Nef (detected by 3D12 or EH1) may represent defective proviruses. These considerations highlight the potential of this assay in future studies aimed at assessing the HIV reservoir in patients. Additionally, these immunofluorescence strategies are broadly applicable to other proteins, as long as suitable antibodies are available to distinguish different expression patterns of the target of interest, therefore, providing a more comprehensive assessment than traditional methods.

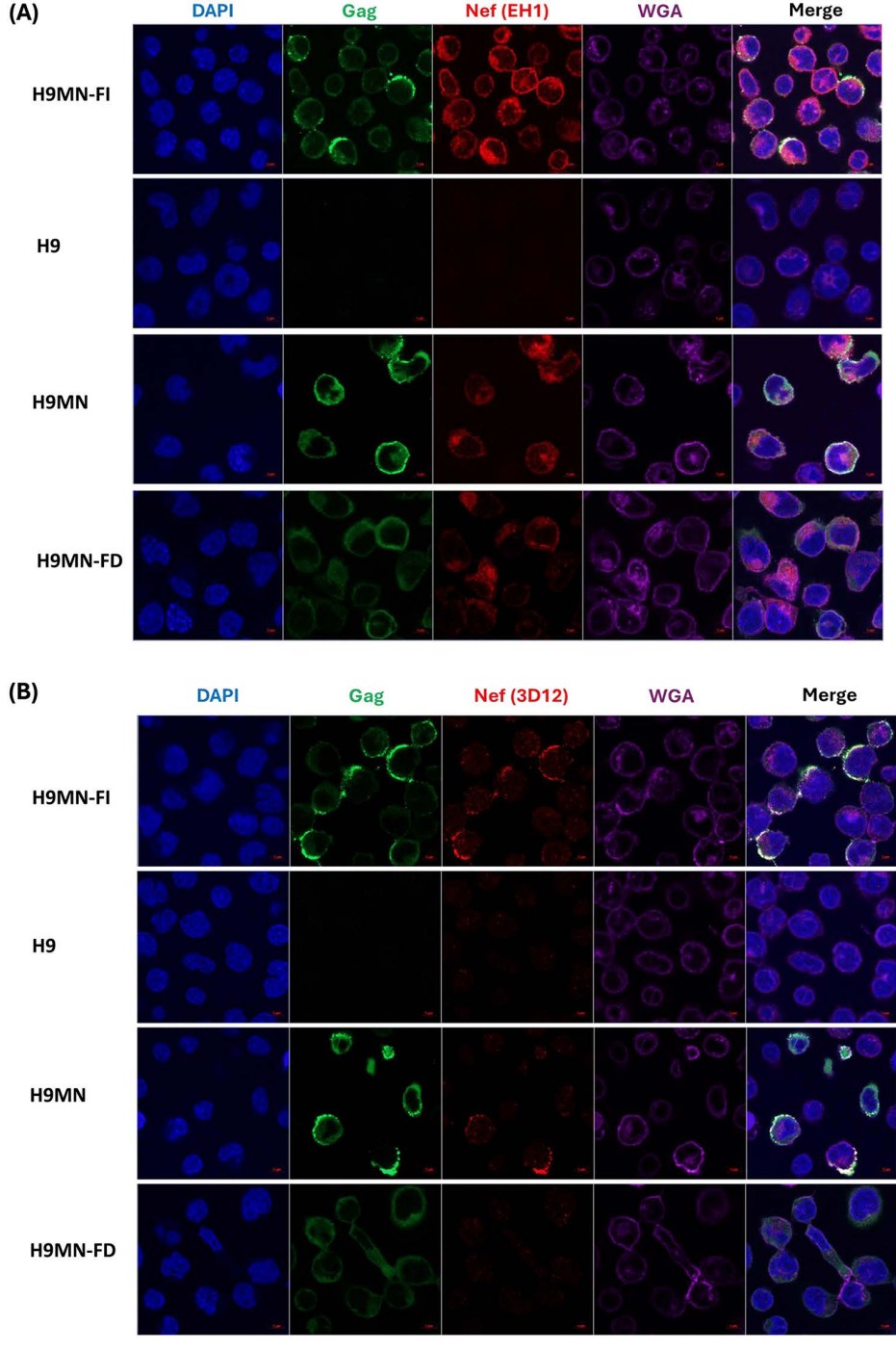

**Fig 3. Confocal microscopy images comparing Gag and Nef (clones EH1 and 3D12) expression in H9MN and H9MN-FD cells.** For the confocal images, the following cellular components were stained: the nuclei (blue), Gag protein (green), Nef protein (red) and the plasma membrane (purple). Scale bar: 5 μm. **A)** Dual screening with Nef (EH1) and Gag antibodies. **B)** Dual screening with Nef (3D12) and Gag antibodies.

## Supporting information

**S1 File. Step-by-step protocol, also available on protocols.io.**
(PDF)

**S1 Table. A list of primary antibodies screened in the study.**
(PDF)

## Acknowledgments

The following reagent was obtained through the NIH HIV Reagent Program, Division of AIDS, NIAID, NIH: Human Immunodeficiency Virus 1 (HIV-1) MN-Infected H9 Cells, ARP-402, contributed by Dr. Robert Gallo; Monoclonal Anti-Human Immunodeficiency Virus Type 1 (HIV-1) Nef Protein (EH1), ARP-3689, contributed by Dr. James Hoxie. All illustrations and workflows were created using BioRender.com.

## Author contributions

**Conceptualization:** Hiromi Imamichi, H Clifford Lane, Tomozumi Imamichi.

**Data curation:** Rosana Wiscovitch-Russo, Hongyan Sui.

**Formal analysis:** Hongyan Sui.

**Funding acquisition:** Tomozumi Imamichi.

**Investigation:** Rosana Wiscovitch-Russo, Hongyan Sui.

**Methodology:** Hongyan Sui.

**Project administration:** Hongyan Sui, Hiromi Imamichi, H Clifford Lane, Tomozumi Imamichi.

**Resources:** Mindy Smith, Hiromi Imamichi.

**Supervision:** Hongyan Sui, H Clifford Lane, Tomozumi Imamichi.

**Validation:** Rosana Wiscovitch-Russo, Hongyan Sui.

**Writing – original draft:** Rosana Wiscovitch-Russo, Hongyan Sui.

**Writing – review & editing:** Hongyan Sui, Hiromi Imamichi, H Clifford Lane, Tomozumi Imamichi.

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
