## [Decision Letter · Decision Letter 0]

10 Nov 2025

An immunofluorescence microscopy assay to discriminate distinct expression patterns of HIV-1 Gag and Nef proteins in HIV-1 provirus-harboring cells

PLOS ONE

Dear Dr. Sui,

Thank you for submitting your manuscript to PLOS ONE. After careful consideration, we feel that it has merit but does not fully meet PLOS ONE’s publication criteria as it currently stands. Therefore, we invite you to submit a revised version of the manuscript that addresses the points raised during the review process.

We look forward to receiving your revised manuscript.

Kind regards,

Michael Schindler

Academic Editor

PLOS ONE

Journal Requirements:

 “This project has been funded in whole or in part with federal funds from the National Cancer Institute, National Institutes of Health, under contract number HHSN261200800001E.” 

“The following reagent was obtained through the NIH HIV Reagent Program, Division of 189 AIDS, NIAID, NIH: Human Immunodeficiency Virus 1 (HIV-1) MN-Infected H9 Cells, ARP-402, 190 contributed by Dr. Robert Gallo; Monoclonal Anti-Human Immunodeficiency Virus Type 1 (HIV191 1) Nef Protein (EH1), ARP-3689, contributed by Dr. James Hoxie. All illustrations and workflows 192 were created using BioRender.com. This project has been funded in whole or in part with 193 federal funds from the National Cancer Institute, National Institutes of Health, under contract 194 number HHSN261200800001E. This research was supported in part by the Intramural Research 195 Program of the National Institutes of Health (NIH). The contributions of the Mindy Smith, Hiromi 196 Imamichi, and H. Clifford Lane are considered Works of the United States Government. The 197 findings and conclusions presented in this paper are those of the authors and do not necessarily 198 reflect the views of the NIH or the U.S. Department of Health and Human Services.”

“This project has been funded in whole or in part with federal funds from the National Cancer Institute, National Institutes of Health, under contract number HHSN261200800001E.” 

5. We note you have not yet provided a protocols.io PDF version of your protocol and/or a protocols.io DOI. When you submit your revision, please provide a PDF version of your protocol as generated by protocols.io (the file will have the protocols.io logo in the upper right corner of the first page) as a Supporting Information file. The filename should be S1_file.pdf, and you should enter “S1 File” into the Description field. Any additional protocols should be numbered S2, S3, and so on. Please also follow the instructions for Supporting Information captions [https://journals.plos.org/plosone/s/supporting-information#loc-captions]. The title in the caption should read: “Step-by-step protocol, also available on protocols.io.”

Please assign your protocol a protocols.io DOI, if you have not already done so, and include the following line in the Materials and Methods section of your manuscript: “The protocol described in this peer-reviewed article is published on protocols.io (https://dx.doi.org/10.17504/protocols.io.[...]) and is included for printing purposes as S1 File.” You should also supply the DOI in the Protocols.io DOI field of the submission form when you submit your revision.

If you have not yet uploaded your protocol to protocols.io, you are invited to use the platform’s protocol entry service [https://www.protocols.io/we-enter-protocols] for doing so, at no charge. Through this service, the team at protocols.io will enter your protocol for you and format it in a way that takes advantage of the platform’s features. When submitting your protocol to the protocol entry service please include the customer code PLOS2022 in the Note field and indicate that your protocol is associated with a PLOS ONE Lab Protocol Submission. You should also include the title and manuscript number of your PLOS ONE submission.

Additional Editor Comments:

Your manuscript was reviewed by two experts and there is agreement your protocol has some merit. However, revisions explicitly as stated by reviewer 1 are necessary. Especially, you should present the manuscript strictly in a style that is adequate for a protocol. 

Reviewer's Responses to Questions

**Comments to the Author**



Reviewer #1: No

Reviewer #2: Yes

2. Has the protocol been described in sufficient detail?

To answer this question, please click the link to protocols.io in the Materials and Methods section of the manuscript (if a link has been provided) or consult the step-by-step protocol in the Supporting Information files.

Reviewer #1: No

Reviewer #2: Yes

3. Does the protocol describe a validated method?

Reviewer #1: Yes

Reviewer #2: Yes

4. If the manuscript contains new data, have the authors made this data fully available?

Reviewer #1: Yes

Reviewer #2: Yes

**5. Is the article presented in an intelligible fashion and written in standard English?**

Reviewer #1: Yes

Reviewer #2: Yes

Reviewer #1: Wiscovitch-Russo et al. set out to develop an immunofluorescence assay that identifies Nef and Gag in cells containing HIV proviruses. After determining the optimal order for WGA staining – prior to fixation – for identification of the plasma membrane, they screened fourteen anti-Nef antibodies and four anti-HIV-1 p24 antibodies. The anti-Nef clones EH1 and 3D12 were selected and subsequently utilized in cells containing intact proviruses (H9MN-FI), a mixture of intact and defective proviruses (H9MN), and only a single defective proviral clone (H9MN-FD). H9 cells containing no proviruses served as a negative control. Consistently, the antibody EH1 clone was associated with cytoplasmic Nef expression, while the 3D12 clone signal was membrane localized. Notably, Nef in H9MN-FD cells did not localize at the plasma membrane, indicating that proteins translated from defective proviruses may be altered from their intact-proviral derived counterparts.

The following revisions would improve the manuscript.

1. The authors have presented a manuscript as a “Protocol”, yet most of the methods are in supplemental materials. This information should be in the main text for this type of format.

2. The authors should more clearly explain why they tested EH1 and 3D12 anti-Nef antibody clones in the intact provirus H9s (H9MN-FI).

3. Pearson Correlation plots: These correlation plots are from analysis of individual cells/images – since the authors do not disclose how many cells they analyzed per condition, I’m assuming they analyzed the cells within the region of interest depicted in their images (Fig 2 C & D) – could benefit from clarification

o Again for the Pearson Correlation plots, a bigger sample size should be included; averaging out the PCC value will give more validity to their results

• For Figure 3, the authors should also perform the colocalization analysis with the H9MN cells with the mixed intact and defective proviral clone populations, and the H9MN-FD cells with only defective proviruses – to bolster their observations that there are differences in Nef localization

4. For immunostaining protocols, the authors should include antibody dilution info.

5. For the Imaging and Analysis:

• What plug-in within Fiji was used to do the colocalization analysis? (either Coloc 2 or JACoP)?

• The authors should define what metrics they are using to quantify colocalization (i.e. PCC, MOC)

• Need to include the software versions they utilized – for both ZEN and Fiji

6. For the quantification via Nef and WGA stain colocalization analysis:

Could the lack of membrane localization in the H9MN-FD condition have to do with some alteration to the N-terminal myristoylation

Minor Comments :

Line 40: This latent reservoir is a major barrier to a cure

Line 43: ‘Intact’ should not be capitalized

Line 48/49: Replace ‘HIV-infected individuals’ with ‘people living with HIV-1’

Line 52: No need to define Group-specific antigen and Negative regulatory factor –

Line 54: Switch order of ‘accessory multifunctional’ → multifunctional accessory protein

Line 56: Already mentioned that Nef is an accessory protein in the previous sentence; would remove ‘is an auxiliary protein’

Line 76: Verb tense change – should be ‘reached’ instead of reach

Line 145: ‘It’ should be lowercase

Reviewer #2: Overview: This is a timely and interesting methods paper given the emerging picture of the clinical relevance of defective proviruses in the HIV-1 latent reservoir. The paper describes a staining protocol using two antibodies to HIV-1 Nef that can differentiate proteins produced from intact and defective HIV-1 proviruses in combination with an anti-HIV-1 Gag antibody. The manuscript is concise but clear and was a pleasure to read. The protocol is clear and easy to understand with a good level of detail for replicating the experiments in other settings. Most of the points in the revision document are recommend small changes (like typos) to the manuscript and supplementary information. However, point 10 is important to note for translation of this protocol into one that can be used with clinical samples.

**Do you want your identity to be public for this peer review?** For information about this choice, including consent withdrawal, please see our Privacy Policy

Reviewer #1: No

Reviewer #2: No

---

## [Author Response · Author response to Decision Letter 1]

12 Dec 2025

Please check attached "Response to reviewers" for details.

---

## [Decision Letter · Decision Letter 1]

22 Dec 2025

An immunofluorescence microscopy assay to discriminate distinct expression patterns of HIV-1 Gag and Nef proteins in HIV-1 provirus-harboring cells

PONE-D-25-53504R1

Dear Dr. Sui,

We’re pleased to inform you that your manuscript has been judged scientifically suitable for publication and will be formally accepted for publication once it meets all outstanding technical requirements.

Kind regards,

Michael Schindler

Academic Editor

PLOS One

Additional Editor Comments (optional):

Reviewers' comments:

Reviewer's Responses to Questions

**Comments to the Author**



Reviewer #1: Yes

2. Has the protocol been described in sufficient detail?

To answer this question, please click the link to protocols.io in the Materials and Methods section of the manuscript (if a link has been provided) or consult the step-by-step protocol in the Supporting Information files.

Reviewer #1: Yes

3. Does the protocol describe a validated method?

Reviewer #1: Yes

4. If the manuscript contains new data, have the authors made this data fully available?

Reviewer #1: Yes

**5. Is the article presented in an intelligible fashion and written in standard English?**

Reviewer #1: Yes

Reviewer #1: The authors have addressed the concerns and comments that were raised in the first round of reviews.

**Do you want your identity to be public for this peer review?** For information about this choice, including consent withdrawal, please see our Privacy Policy

Reviewer #1: No

---

## [Editor Report · Acceptance letter]

PONE-D-25-53504R1

PLOS One

Dear Dr. Sui,

I'm pleased to inform you that your manuscript has been deemed suitable for publication in PLOS One. Congratulations! Your manuscript is now being handed over to our production team.

Kind regards,

on behalf of

Prof. Dr. Michael Schindler

Academic Editor

PLOS One